# The influence of anthropogenic regulation and evaporite dissolution on earthquake-triggered ground failure

Paula Bürgi [1] ✉, Eric M. Thompson [1], Kate E. Allstadt [1], Kyle D. Murray [2], H. Benjamin Mason[1,3], Sean K. Ahdi [4] & Devin Katzenstein[5]

Remote sensing observations of Searles Lake following the 2019 moment magnitude 7.1 Ridgecrest, California, earthquake reveal an area where surface ejecta is arranged in a repeating hexagonal pattern that is collocated with a solution-mining operation. By analyzing geologic and geotechnical data, here we show that the hexagonal surface ejecta is likely not a result of liquefaction. Instead, we propose dissolution cavity collapse (DCC) as an alternative driving mechanism. We support this theory with pre-event Interferometric Synthetic Aperture Radar data, which reveals differential subsidence patterns and the creation of subsurface void space. We also find that DCC is likely triggered at a lower shaking threshold than classical liquefaction. This and other unknown mechanisms can masquerade as liquefaction, introducing bias into liquefaction prediction models that rely on liquefaction inventories. This paper also highlights the opportunities and drawbacks of using remote sensing data to disentangle the complex factors that influence earthquake-triggered ground failure.

On July 6th, 2019, the moment magnitude ($M_w$) 7.1 Ridgecrest, California[1,2], earthquake caused strong shaking, with peak ground acceleration (PGA) between 0.2 and 0.5 g[3], and widespread surface ejecta in the nearby Searles Lake, a semi-dry lakebed ~25 km east of the earthquake epicenter[2,4] (Fig. 1a, b). We define surface ejecta as any subsurface material that migrates upward and is deposited on the surface. Post-earthquake field observations by Zimmaro et al. [1] confirmed the presence of surface ejecta around the periphery and in the southern portion of the lakebed, and those authors attributed the surface ejecta to earthquake-induced liquefaction. Liquefaction occurs when loosely packed, saturated sediments, commonly sand, suddenly contract due to shaking, causing temporarily elevated pore pressures that reduce soil strength[5,6]. Liquefaction is often assumed to be the driving mechanism that causes earthquake-triggered surface ejecta as the elevated pore pressures drive sediment-rich fluids towards the surface[7].

This study focuses on the northeast portion of Searles Lake. Here, surface ejecta is visible in optical satellite imagery and are expressed in a repeating pattern of ~160-m-wide hexagonal cells, hereafter referred to as the hexagonal ejecta (Fig. 1c, d). The hexagonal ejecta are collocated with a pre-existing hexagonal network of fluid injection wells that are part of a local mining operation. Our goal is to understand the processes and mechanisms that led to the hexagonal ejecta.

Liquefaction contributes substantially to earthquake damage and loss, and is commonly considered to be the driving mechanism behind earthquake-triggered surface ejecta. Nevertheless, liquefaction and other ground failure modes remain a challenging process to model at high spatial resolution, over large regions, and through time[8,9]. To improve our ability to predict where and at what thresholds ground failure may occur, we need a more holistic understanding of alternative processes that could lead to surface ejecta.

[1]U.S. Geological Survey Geologic Hazards Science Center, Golden, CO, USA. [2]Department of Earth Sciences, University of Hawaii, Honolulu, HI, USA. [3]College of Engineering, Oregon State University, Corvallis, OR, USA. [4]AECOM, Los Angeles, CA, USA. [5]California Department of Transportation, Sacramento, CA, USA. ✉e-mail: pburgi@usgs.gov

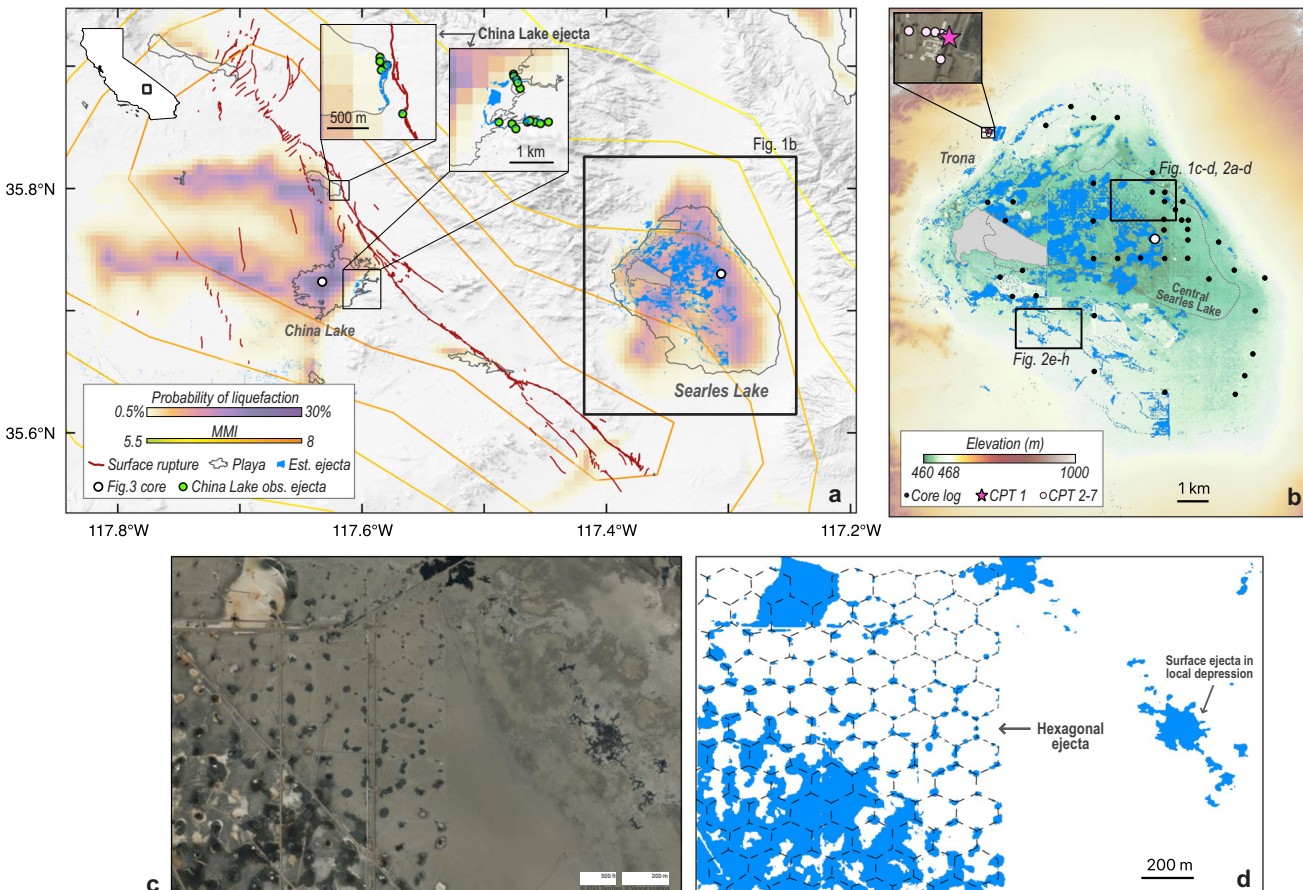

**Fig. 1 | Ground failure triggered by the $M_w$ 7.1 2019 Ridgecrest, California, earthquake. a** Regional context, showing the U.S. Geological Survey (USGS) liquefaction prediction model and ShakeMap contours[3], optically derived change detection using Sentinel-2 data (i.e., estimated ejecta), fault surface rupture[4], and present-day playas[64]. The dots show locations of the two geologic cores shown in Fig. 3a [23]. Insets show two areas with surface ejecta in more detail, with field observation of ejecta (i.e., observed ejecta) shown as green dots. **b** Surface ejecta in Searles Lake (blue) overlying a lidar digital elevation model[65]. Also shown are the locations of USGS geologic cores[20], geotechnical CPT data (enlarged in the inset), and the central portion of the lakebed where the thickest evaporite units are located[10] (dashed line). **c** Optical imagery of the hexagonal ejecta the day after the Ridgecrest earthquake (Bing™Maps, 2023). **d** Optically detected surface change following the Ridgecrest earthquake, generated using © 2019 Planet Labs PBC imagery (refer to "Methods" section). Dashed lines outline the hexagonal ejecta. Note that CPT 1 in **b** is the location of the cone penetration test sounding shown in Fig. 3. Microsoft product screen shot(s) reprinted with permission from Microsoft Corporation.

To understand the processes that led to the hexagonal ejecta observed in Searles Lake, we integrate geologic, geotechnical, and remote sensing data prior to and spanning the 2019 Ridgecrest earthquake sequence. First, we determine whether liquefaction was responsible for the observed surface ejecta in Searles Lake. Using pre-existing geologic cores and geotechnical cone penetration test (CPT) logs to constrain the subsurface composition and expected behavior, we find that liquefaction is likely not the mechanism that triggered the hexagonal ejecta. Next, we present an alternative mechanism for the observed surface ejecta by integrating Interferometric Synthetic Aperture Radar (InSAR) displacement time series results with knowledge about the mining operation. Finally, we compare the observations of surface ejecta in Searles Lake to other surface ejecta observations triggered by the Ridgecrest and other earthquakes, and discuss the implications of different driving mechanisms.

## Results and discussion
### Background
Searles Lake is a present-day playa in the western U.S. Basin and Range province. During the wetter climate conditions of Pleistocene-era glacial periods, Searles Lake held year-round standing water and was the terminal lake in the Owens River system[10]. Searles Lake stratigraphy consists of fine-grained lacustrine units deposited during glacial periods and evaporite units that precipitated out of drying lake waters at the end of glacial periods, when the climate in this region transitioned from temperate to arid[10]. Outside the central lakebed (outlined in Fig. 1b), alluvial deposits are interbedded with the evaporite and lacustrine units.

Since the late 1800s, a variety of methods have been used to extract evaporite mineral products from Searles Lake, including soda ash, borax, and potash[11]. One such method, called in situ leach mining or warm solution mining, manifests as a laterally repeating hexagonal pattern of boreholes. The hexagon vertices are fluid injection wells, which inject warm, chemically unsaturated fluid at ~20 m depth. The injection dissolves evaporite minerals and becomes brine. Brine is pumped out of the subsurface at extraction wells, which are located at the center of each hexagonal cell of injection wells.

Today, the only perennial standing water in Searles Lake are mining-related wastewater ponds. Elsewhere, Searles Lake is covered by dark-colored clay and white halite deposits. The topographic relief within the lakebed is only 2–3 m, including natural and mining-induced local depressions with diameters ranging from meters to hundreds of meters (Fig. 1b). The white-colored surficial halite deposits become visible within the local depressions when they are dry in the weeks to months after a wetting event (i.e., natural precipitation or mining activity). When standing water collects in local depressions due to a

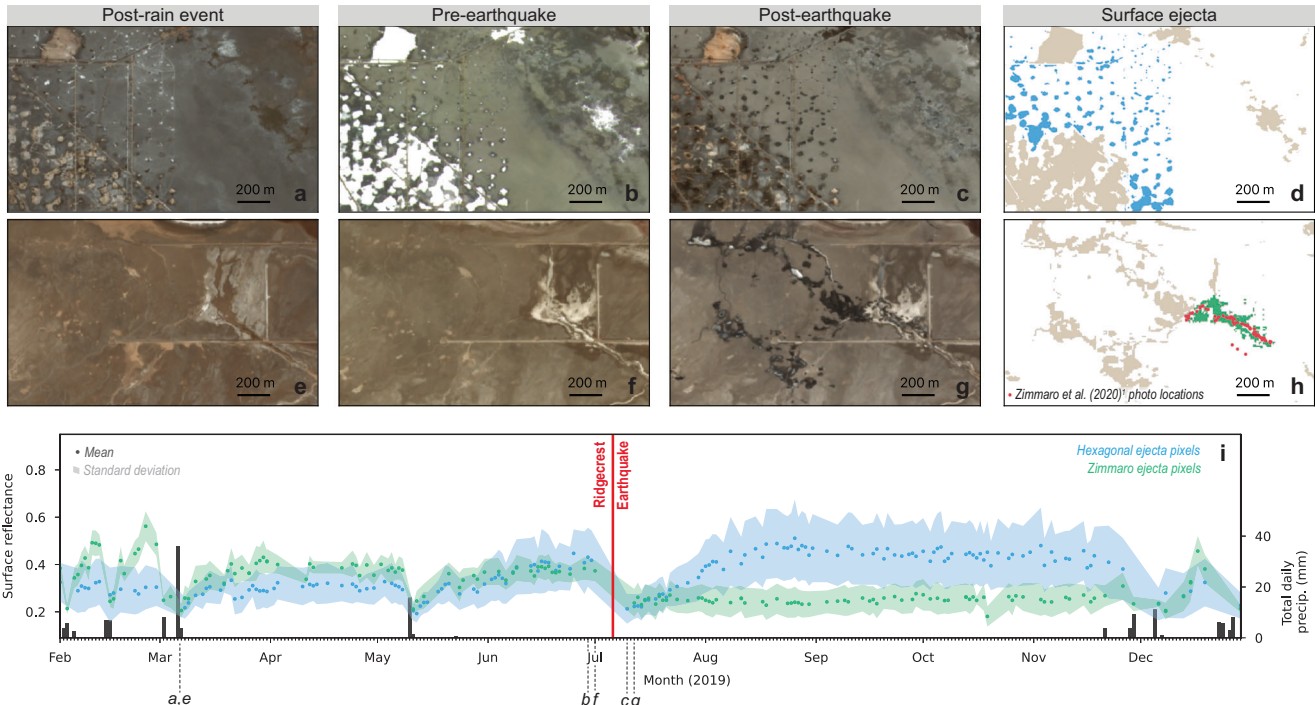

**Fig. 2 | Temporal evolution of optical surface reflectance. a–d** is an area in the vicinity of the hexagonal ejecta (same footprint as Fig. 1c, d) and (**e–h**) is an area containing field-confirmed liquefaction-driven surface ejecta. **a–c**, **e–g** Optical imagery from three dates illustrating differences in surface reflectivity (**a**), (**e**) after a rain event, (**b**), (**f**) after a dry spell and before the earthquake, and (**c**), (**g**) following the earthquake (© 2019 Planet Labs PBC). **d** Blue areas and (**h**) green areas indicate pixels for which surface ejecta are collocated with injection wells and with field-confirmed liquefaction, respectively. **i** Surface reflectance over time for blue and green surface ejecta pixels shown in **d** and **h**. Points indicate the mean surface reflectance value and shaded areas indicates the standard deviation (refer to "Methods" section for details). Note the decrease in overall reflectance both following rain events (right y-axis) and the Ridgecrest earthquake, before which no rain was recorded.

wetting event, the halite deposits within these local depressions dissolve and the surface develops a brown to copper color (Fig. 2a). Over the following weeks as the standing water evaporates and/or percolates into the subsurface, halite precipitates out of solution and the surface returns to a white color (Fig. 2b). In flat areas (i.e., outside of local depressions), brown-to-beige-colored clay deposits remain relatively constant through time.

## Surface ejecta observations

Surface ejecta triggered by the 2019 Ridgecrest Earthquake in Searles Lake was observed with optical remote sensing (refer to "Methods" section) and in the field by mine employees and researchers[1]. Some surface ejecta was also observed after the $M_w$ 6.4 foreshock two days prior, but satellite data show that most of the ejecta occurred in response to the $M_w$ 7.1 mainshock. Field research observations were acquired in November 2019, four months after the earthquake. Although field reconnaissance was limited due to accessibility, the team did observe sand boils (surface ejecta features indicative of liquefaction) in an ephemeral fluvial system at the south end of Searles Lake, approximately 6 km south of the hexagonal ejecta (Fig. 2g). Although the exact source lithologic unit(s) for the field-confirmed observed liquefaction features is unknown, mining-related drilling confirms shallow (<20 m) sand-dominated units immediately surrounding the ephemeral fluvial system.

The sand boil ejection features identified by Zimmaro et al. [1] express a substantial and sudden decrease in reflectivity in optical remote sensing imagery from the pre-event to post-event data. A similar decrease in reflectivity occurred in other parts of the lakebed that the Zimmaro et al. [1] field team could not access, including the hexagonal ejecta (Fig. 2). Rainfall can also decrease surface reflectivity in optical data; however, in this case, it is unlikely that rainfall caused

the change in reflectivity because of the absence of any rainfall in the weeks preceding and following the earthquake (Fig. 2i). Thus, we attribute any pixel that exhibits a characteristic change in reflectivity (refer to Methods) to surface ejecta triggered by the earthquake.

In the months following the earthquake, during which no rainfall was recorded, the hexagonal ejecta and the field-observed sand boils showed distinctly different reflectance behaviors. This trend is illustrated in Fig. 2i, where the pixels containing hexagonal ejecta develop high surface reflectivity in the post-earthquake period. In contrast, the pixels containing field-observed sand boils remain at a lower level of reflectivity in the post-earthquake period. The higher reflectivity of the hexagonal ejecta in the post-earthquake period is likely due to the precipitation of halite out of a brine-rich fluid. Thus, despite their similar reflectivity immediately following the earthquake, this indicates that the composition of the hexagonal ejecta is fundamentally different than the sand boil ejecta observed by Zimmaro et al. [1].

## Anthropogenic influence on Searles Lake surface ejecta

The collocation of the surface ejecta with mining-related wells indicates that mining operations in Searles Lake directly influenced surface manifestation of ejecta. Previous work has shown that land use and human activity can generate conditions conducive to earthquake-triggered ground failure; for instance, liquefaction commonly occurs in artificial fill, which is a result of human activity[12,13]. Land use related to agricultural irrigation, which artificially raises the water table, can also lead to liquefaction in shallow layers[14–16]. To our knowledge, the earthquake-triggered surface ejecta presented here is the first documented case directly related to mining activity. In addition, the stark correlation between the mining-related wells and surface ejecta indicate that the wells provided a conduit for the pressurized sediment and fluid to escape, which provides more documented evidence of

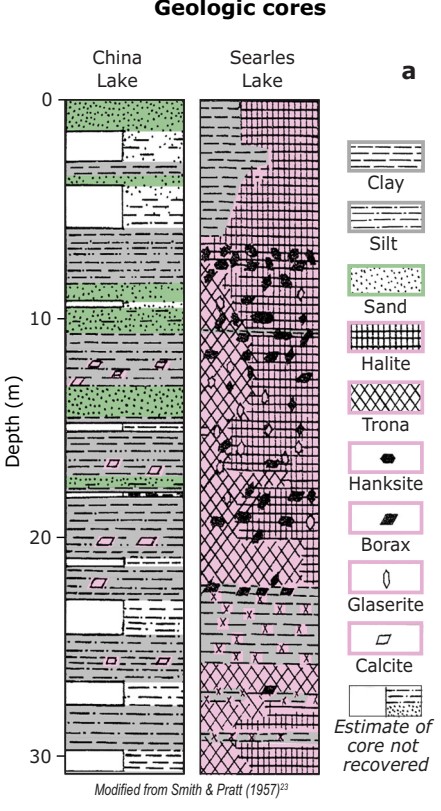

## Geologic cores
China Lake    Searles Lake

**a**

Clay
Silt
Sand
Halite
Trona
Hanksite
Borax
Glaserite
Calcite
*Estimate of core not recovered*

*Modified from Smith & Pratt (1957)[23]*

## Geotechnical CPT data

**b** Normalized cone resistance, Qtn

Normalized friction ratio, Fr (%)

**c** Normalized friction ratio, Fr (%)
$I_c = 2.6$

Normalized cone resistance, Qtn

SD, TD, CD, SC, TC, CCS, CC

**d**

% Total SBT

CCS 23.0% · CC 55.6% · CD 0.4% · TC 2.4% · TD 1.3% · SC 2.2% · SD 15.1%

SBT

**Fig. 3 | Subsurface geological and geotechnical data. a** Example geologic cores acquired in Searles Lake and China Lake. Smith & Pratt[23] interpret the repeating pattern symbols (i.e., clay, silt, sand, halite, and trona) to represent their relative abundance within a given layer, whereas the presence of stand-alone mineral symbols (i.e., trona, hanksite, borax, glaserite, and calcite) indicate the presence, but not necessarily the proportional abundance, of the mineral within a given layer. Note absence of any sand-sized particles in the Searles Lake core. **b** Example CPT log (CPT-1), location shown in Fig. 1b. Fr and Qtn are used to deduce SBT, which correlate with liquefaction susceptibility[30]. SBT acronyms are defined in Supplementary Table S1. Materials considered to be classically liquefiable (SBT type SC) are shaded green in **b** and colored green in **c, d**. Materials that fall within the other six categories are shaded in gray in (b) and colored gray in **c, d. c** Fr versus Qtn at each depth interval for all CPT logs (CPT log locations are shown in Fig. 1b, inset). Black lines delineate between different SBTs, and red dashed line shows the $I_c = 2.6$ isoline. **d** Bar chart showing the relative abundance of each SBT for all CPT logs. Note that these CPT logs were acquired on the periphery of the lakebed, which is topped by an alluvial cover not present on the central lakebed (alluvial portion of CPT data is shown as thin dashed lines and no shading in **b** and not included in **c, d**).

direct anthropogenic regulation of the surface manifestation of subsurface ground failure.

### Was liquefaction responsible for the hexagonal ejecta?

Having established the correlation between the hexagonal ejecta and anthropogenic activity, our focus now shifts to an examination of the driving mechanism that caused such extensive surface ejecta. Our initial hypothesis was that the hexagonal ejecta resulted from liquefaction. This is supported by three lines of reasoning: (1) there is striking similarity between the field-confirmed liquefaction features in southern Searles Lake[1] and the hexagonal ejecta in optical imagery, (2) liquefaction is recognized as the most common driving mechanism of earthquake-triggered surface ejecta[17], and (3) surface ejecta and injection wells are collocated, and the wells appear to have provided a conduit for liquefied material to escape.

Liquefaction is one of the most complicated and controversial subjects in geotechnical engineering[17], thus we provide an authoritative definition (heretofore referred to as classical liquefaction): "Liquefaction is the transformation of a granular material from a solid to a liquefied state as a consequence of increased pore-water pressure and reduced effective stress"[18].

Liquefaction has been predominantly observed in sand-sized granular materials, while similar behavior in fine-grained (i.e., silts and clays) materials is more complicated. Boulanger & Idriss[19] recommended that the term liquefaction be used for fine-grained material that exhibits sand-like behavior (which in a loose, saturated state tends to contract and undergo liquefaction), while the term cyclic softening can be used for soils that exhibit clay-like behavior that would not be expected to undergo classical liquefaction[19]. Here, we review the available geologic and geotechnical data to understand the mechanism that resulted in surface ejecta, given an initial expectation that surface ejecta typically results from liquefaction.

Because liquefaction occurs primarily in sand-dominated layers, we expect to find that Searles Lake strata contain sand-rich units. Thus, we used geologic cores from Searles Lake to constrain grain size in the near surface. We analyzed dozens of published geologic cores that have been collected over the last century, owing to Searles Lake's long-recognized economic potential[11,20–23] (refer to Fig. 1b for core locations). The vertical extent of the geologic cores ranged from 10 to 30 m deep. Figure 3a shows a representative geologic core acquired in Searles Lake approximately 1 km south of the hexagonal ejecta, and a representative geologic core from China Lake, a dry lakebed approximately 25 km west of Searles Lake and through which the Ridgecrest earthquake fault ruptured[23] (locations in Fig. 1a). The Searles Lake core shows that the central lakebed stratigraphy is composed of clay, silt, and evaporites and notably lacks any sand-sized

sediments. The extensive subsurface geologic mapping in Searles Lake reveals that the core log in Fig. 3a is representative of the central portion of the lakebed (inclusive of the hexagonal ejecta, outlined in Fig. 1a), with only small differences in unit thicknesses and sequence variations and no sand-sized particles in non-evaporite units[20,21,23]. Logs closer to the periphery of the lakebed document fewer evaporites and more clastic material, including sand-sized particles[20]. Review of the central Searles Lake stratigraphy indicates a lack of sand-dominated layers, contradicting a liquefaction-driven surface ejecta hypothesis.

Although grain size alone is a useful proxy for liquefaction susceptibility, a more nuanced understanding can be provided by assessing soil behavior. CPT data are widely used to estimate soil geotechnical properties and stratigraphy, and thus can be used to quantitatively assess liquefaction susceptibility[24–26]. CPT data collection involves pushing a cone-tipped rod into the subsurface and measuring sleeve friction, tip resistance, and pore water pressure at regular intervals. These parameters are then used to classify the subsurface into different categories related to liquefaction susceptibility[25–31]. We present results using soil behavior type[31] (SBT) and soil behavior index[26] ($I_c$) metrics. Other CPT-based liquefaction analyses are presented in Supplementary Information Section 1, including estimates of the probability of liquefaction manifestation at the surface[29].

We analyzed the seven publicly available CPT logs[32] in the area, which were acquired on the western edge of the lakebed in Trona, California, and shown in Fig. 1b. Figure 3b shows one of the seven CPT logs used in this study, and Fig. 3c shows all seven CPT logs overlaid by the SBT and $I_c$ delineations[26,31]. $I_c$ is a numerical index where $I_c < 2.6$ implies sand-dominated strata, whereas $I_c > 2.6$ implies clay- and silt-dominated strata. Here, $I_c < 2.6$ comprise 18.5% of the log intervals, and $I_c > 2.6$ comprises 81.5%. The bar chart in Fig. 3d shows the total percentage of the depth of the CPT log that falls into each SBT. The SBT most associated with liquefaction is contractive sands (SC); layers that contain contractive sands are highlighted in green in Fig. 3b–d. For the seven CPT logs analyzed here, contractive sands comprise only 2.2% of all sampled layers. Further, these CPT logs are from the periphery of the lakebed (Fig. 1b inset), where geologic data[20] indicate that the non-evaporite units contain more sand-sized particles. Our review of the lakebed-peripheral CPT data further bolsters the notion that lakebed-central strata are likely not susceptible to liquefaction.

The geologic core and CPT-derived soil behavior data do not support our initial hypothesis that liquefaction, as defined above, was the dominant mechanism behind the hexagonal ejecta in Searles Lake. That said, liquefaction cannot be exhaustively ruled out; for example, there may be discontinuous or thin layers of liquefiable material underlying the hexagonal ejecta that is not captured in the geologic cores. However, the available data, along with the severity of the surface manifestation, do not align with the defined concept of classical liquefaction.

In this paper, we use the portmanteau liquefiction for fictional liquefaction. We do not use this term to describe a physical process, but rather to describe the misinterpretation of earthquake-triggered ground failure observations, such as surface ejecta, as being driven by classical liquefaction. Because liquefaction is rarely directly observed, this misinterpretation is a common concern; examples include confusing cyclic softening features in fine-grained soils and indistinct surface changes observed in remotely sensed data. In the next section, following our description of an alternative driving mechanism, we discuss whether the hexagonal ejecta could still be liquefaction-driven under a broader definition of the term.

## An alternative mechanism for earthquake-triggered surface ejecta: dissolution cavity collapse

To inform an alternative mechanism that explains the hexagonal ejecta, we consider the impact of solution mining. The collocation of the surface ejecta with mining-related wells is evidence that the wells

provided a conduit for fluid escape; however, it remains unclear if the mining process itself played a role in generating conditions for ground failure. To address this question, we use InSAR displacement time series data to better understand the location and extent of subsurface disturbance leading up to the Ridgecrest earthquake.

We use Sentinel-1 InSAR data from ascending path 64 (104 scenes) and descending path 71 (100 scenes) to generate an average pre-event velocity map (Fig. 4a) and full displacement time series (Fig. 4c, d) spanning 01/02/2015–07/04/2019 (refer to Methods for details). We do not include data spanning the earthquake because the extensive surface ejecta resulted in a complete loss of a usable InSAR signal (i.e., loss of coherence).

The pre-seismic InSAR velocity map (Fig. 4a) reveals a hexagonal pattern of subsidence where more subsidence generally coincides with fluid injection wells (shown in Fig. 4b). To explore this further, we present the full InSAR time series for different groups of pixels (Fig. 4c, d). In Fig. 4c, we compare the mean displacement over time for pixels in an unmined area (control group) with pixels in mined areas where mining initiated at different times (phases 1 and 2). Phase 1 pixels coincide with surface ejecta and with fluid injection wells drilled pre-2015, before the start of the InSAR time series (Fig. 4b–d, purple), phase 2 pixels coincide with surface ejecta and with fluid injection wells drilled in mid-2017, 2.5 years into InSAR time series (Fig. 4c, d, orange), and control group pixels do not coincide with any mining activity or surface ejecta (Fig. 4b, d, green). In Fig. 4d, we compare the mean displacement for pixels that coincide with phase 1 injection wells versus phase 1 extraction wells.

The control group shows a modest amount of subsidence (<1 cm/year), on par with subsidence rates in nearby dry lakebeds[33]. Phase 1 pixels show a relatively constant subsidence rate of 3–4 cm/year over the course of the time series, which is substantially greater than the <1 cm/year rate of the control group. However, phase 2 pixels show two distinct subsidence epochs; from 2015 to mid-2017, before mining initiates, the subsidence rate resembles the control group. From mid-2017 to mid-2019, after mining initiates, the subsidence rate of phase 2 pixels increases to a similar rate as phase 1 pixels. For all three groups, precipitation has some influence on the InSAR displacements; following precipitation events, there is often a short-lived increase in the subsidence rate. However, the different rates of subsidence between mined and unmined areas are apparent even during dry spells with no precipitation-related subsidence. Additionally, subsidence at one active Searles Lake solution mining injection well was confirmed in the field, supporting the assertion that the InSAR data show real deformation (refer to Supplementary Information Section 2 and Fig. S2 for details).

InSAR data reveal that the ground is subsiding at a higher rate around fluid injection wells than fluid extraction wells and unmined areas (Fig. 4c, d). This observation, together with the previously described process of solution mining, provides an alternative driving mechanism for the hexagonal ejecta, as follows: The fluid introduced into the subsurface via injection wells is chemically unsaturated (i.e., no dissolved evaporite minerals) and dissolves soluble material in the injected layers, which creates cavities in the subsurface. As these cavities slowly compact over time, the overlying ground surface subsides. The subsidence rate at the injection wells in Searles Lake is ~3–5 cm/year, which is comparable with, and in some cases less than, subsidence rates linked to evaporite dissolution[34–40]. Thus, we propose a process, similar to liquefaction, that also results in surface ejecta (Fig. 5): (1) dissolution of evaporites increases the void/cavity space that is filled with fluid, (2) earthquake ground shaking causes void/cavity collapse (i.e., a volume reduction), (3) the collapse increases the fluid pressure, and (4) the increased pressure results in sediment-rich fluid to flow to the surface.

One could make the case that DCC is a sub-type of liquefaction; for example, DCC meets the criteria of increased pore-water pressure

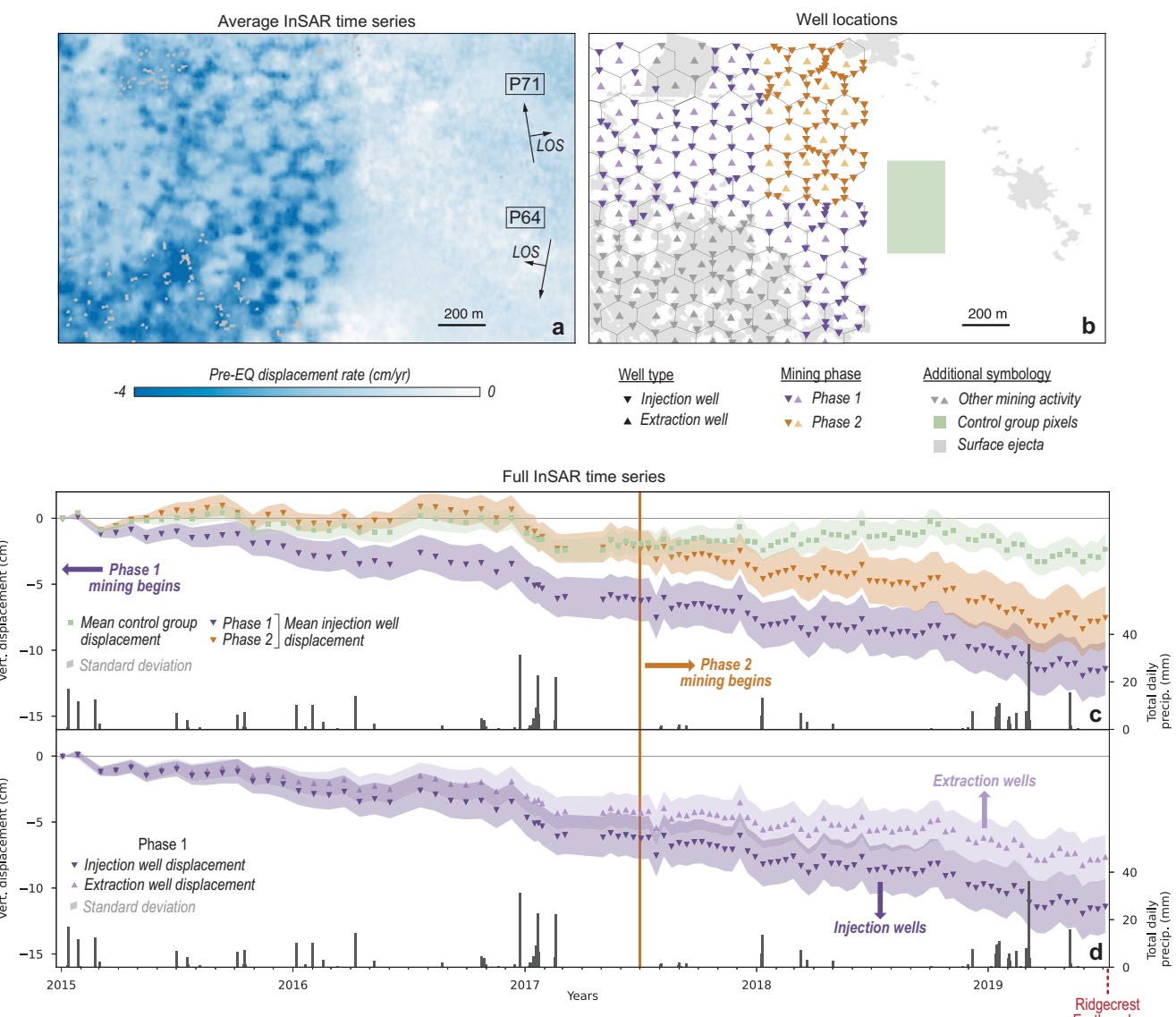

**Fig. 4 | InSAR-derived subsidence before the Ridgecrest earthquake. a** Pre-seismic (01/02/2015–07/04/2019) vertical InSAR displacement rate. Ascending and descending InSAR imaging geometry (i.e., flight direction and line-of-sight (LOS)) is shown on the right. **b** Locations of injection and extraction wells (downward-facing and upward-facing triangles, respectively) relative to surface ejecta (gray areas). The green area indicates the location of the control group pixels, where there is no mining prior to the Ridgecrest earthquake. **c** Full InSAR time series showing vertical displacement for pixels that coincide with the unmined area (green) and the phase 1 and 2 injection wells (dark purple and dark orange, respectively), where symbols indicate the mean value and shaded regions indicate the standard deviation for each category. **d** Full InSAR time series for phase 1 mining injection wells (dark purple) and extraction wells (light purple).

in the definition of classical liquefaction. Furthermore, surface ejecta observations that do not adhere to the full classical definition have previously been attributed to liquefaction. For example, tsunami- or wave-induced liquefaction occurs under high pore pressure conditions, but not necessarily as a result of the contraction of granular material[41]. However, in the opinion of the authors, DCC is fundamentally different than classical liquefaction. We support this assertion in the next section, where we discuss other potential instances of earthquake-triggered DCC and the shaking threshold required for DCC versus classical liquefaction.

### Beyond the hexagonal ejecta

Although warm solution mining is only located in one portion of Searles Lake (i.e., collocated with the hexagonal ejecta), DCC is a viable mechanism for surface ejecta throughout the central

lakebed, in both mined and currently unmined areas. A full analysis of all Searles Lake ejection features is outside the scope of this study, but here we briefly discuss this idea in the context of one surface ejecta feature east of the hexagonal ejecta, in a currently unmined area (location shown in Fig. 1d). This area is one of many local depressions that (1) fall within the central portion of the lakebed where the geological and geotechnical data do not support liquefaction as a dominant process and (2) hosts standing water following rain events[42]. Here, we posit that the ponded evaporite-unsaturated rainwater percolates into the subsurface, which also causes evaporite dissolution and the formation of void space in the shallow subsurface. Rainwater percolation in local depressions is documented in Searles Lake[42], with strong groundwater flow promoting this process[43]. Evaporite dissolution via natural percolation is likely a slower and more stable process over time, possibly

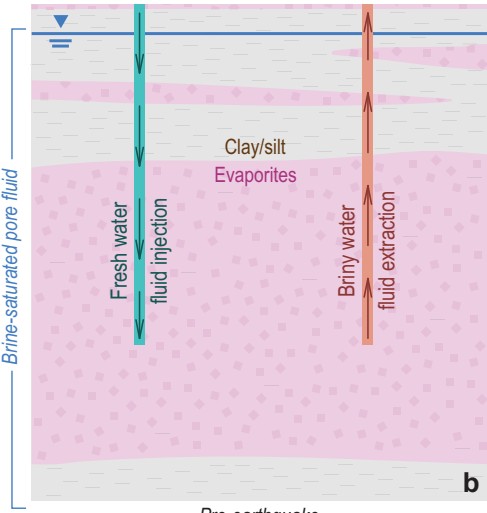

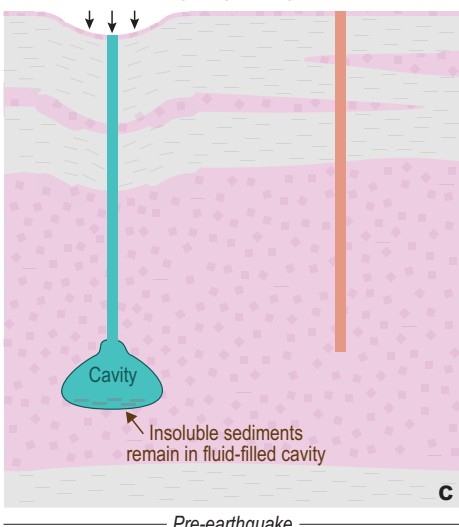

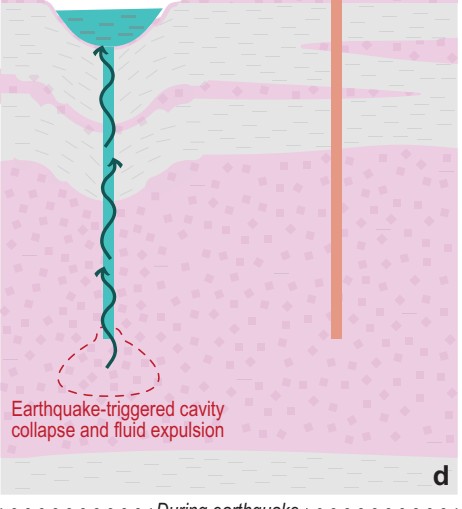

**Fig. 5 | Conceptual illustration of dissolution cavity collapse (DCC). a** Oblique view illustrating the process of warm solution mining (based on Collet et al.)[66]. **b–d** The relationship between mining activity, subsurface disturbance, surface deformation, and earthquake-triggered surface ejecta. Note that this conceptual illustration shows only one possible expression of cavity space; in reality, a variety of cavity manifestations are plausible, ranging from many micro-scale cavities (essentially, an increase in pore space) to a single macro-scale void (as depicted here).

explaining the absence of an elevated pre-seismic InSAR subsidence rate over this local depression.

Next, we consider observations of surface ejecta in other lakebeds impacted by the Ridgecrest earthquake. Despite the extensive DCC-driven surface ejecta in Searles Lake, the neighboring China Lake exhibits notably less ejecta (Fig. 1a). The two insets in Fig. 1a show zoomed-in views of China Lake that did experience ejecta, along with field observations. Due to proximity to the Ridgecrest rupture, China Lake, on average, experienced stronger shaking (PGA of ~0.5 g) compared to Searles Lake (PGA of ~0.35 g)[3] (Fig. 1a). To understand this discrepancy, we compare geologic cores from each lakebed (Fig. 3a). The China Lake core exhibits a range of sand, silt, and clay layers with minimal evaporites, whereas the Searles Lake core records only clays and extensive evaporites. In addition, sand boils were observed in China Lake[2], which are indicative of classical liquefaction. This indicates that classical liquefaction primarily drove surface ejecta in China Lake, whereas DCC

primarily drove surface ejecta in Searles Lake. Because Searles Lake experienced less intense shaking but more extensive ejecta, we conclude that DCC may be triggered at a lower shaking threshold than classical liquefaction.

The ground failure phenomenon observed in Searles Lake may not be unique to this lakebed or earthquake; DCC could occur anywhere with shallow, dissolution-susceptible (e.g., gypsum, carbonate) deposits and a shallow groundwater table. In fact, unusually extensive earthquake-triggered ground failure has been reported in both evaporite-rich and carbonate-rich regions[44–48]. For example, the 2020 Monte Cristo, Nevada, earthquake produced a relatively small degree of shaking within a nearby salt marsh (PGA of ~0.2 g), but caused large circular subsidence features within the salt marsh[47]. In their report, the field reconnaissance team noted the oddity of such features, hypothesizing that "[d]uring the earthquake, the salt crystals were crushed and the ground collapsed," and "[t]he response at the average ground conditions suggest that, with a slightly more powerful earthquake,

significantly more ground failure might have been observed at the Columbus Salt Marsh site"[47].

The 2001 Gujarāt, India (or Bhuj) earthquake is another event that caused extensive ground failure and surface ejecta features within a large semi-dry salt flat[48]. Field teams observed mud volcanoes notably lacking traditionally liquefiable material (i.e., sand-dominated), and optical satellite observations revealed extensive dark-colored ejecta that, over the next weeks to months, turned white due to evaporite minerals precipitating out of the ejected fluid[48]. The sheer amount of ejecta throughout the salt flat, fines-dominated ejected material, and dark-to-light color change of the ejecta post-event (Fig. 2i) is all strikingly similar to what we observed in Searles Lake. Thus, we hypothesize that much of the Gujarāt earthquake-triggered ground failure attributed at the time to liquefaction was, in fact, DCC.

## Implications

Based on the evidence above, DCC appears to be neither unique to Searles Lake nor exclusively a result of anthropogenic activity. There is a strong likelihood that similar occurrences have been witnessed but not explicitly named in other earthquake events, such as the Monte Cristo, Nevada, earthquake and the Gujarāt/Bhuj earthquake (described above), the latter of which caused extensive damage and human fatalities[49]. Furthermore, substantial portions of land worldwide are underlain by evaporite deposits[50], including 40% of the contiguous United States[51]. These examples highlight the elevated risk that may be posed by DCC in earthquake events.

The extensive surface ejecta likely caused by DCC also appears to be triggered at a lower shaking threshold compared to classical liquefaction. DCC observations are likely included in global liquefaction inventories[52], which are a key input to liquefaction prediction models[53]. Consequently, the inclusion of these observations in liquefaction models may introduce bias, leading to an overprediction of liquefaction and an underprediction of DCC. This highlights the importance of recognizing and accounting for the distinct characteristics of DCC and other liquefiction ground failure mechanisms in seismic hazard assessments.

Optical remote sensing observations of the 2019 Ridgecrest, California, earthquake revealed extensive surface ejecta in Searles Lake, including one area where the surface ejecta was arranged in a repeating hexagonal pattern. The hexagonal ejecta are collocated with injection wells from a solution-mining operation, reiterating how anthropogenic activities can be a primary controller of the spatial distribution of surface ejecta. Anthropogenic land generation (i.e., artificial fill) is well known to be susceptible to earthquake-triggered ground failure[12,13], but this study shows that land modification and cavities created by fluid flow are also important factors in ground failure susceptibility.

The surface ejecta in Searles Lake could easily be interpreted as liquefaction using remote sensing data; however, geologic and geotechnical data indicate that central Searles Lake deposits are unlikely to host appreciable liquefiable material, and we conclude that classical liquefaction was not the dominant mechanism of ground failure. Preseismic InSAR data revealed substantial (>3 cm/year) subsidence at injection wells; this observation led us to hypothesize that the observed surface ejecta was a result of shaking-induced dissolution cavity collapse (DCC) generated by the dissolution of evaporite-rich layers in the subsurface. Although evaporite dissolution was driven in large part by local mining operations, the presence of extensive surface ejecta in non-mined local depressions throughout the lakebed indicates that this process can occur naturally. We also find that DCC may be triggered at a lower shaking threshold than classical liquefaction.

With the advent of frequently acquired high-resolution remote sensing data, we show that earthquake-triggered ground failure research can gain a new perspective by comprehensively interrogating the spatial relationships between surface manifestation and surface features, subsurface properties, and land use. However, this study also acts as a cautionary tale: remote sensing methods excel at detecting large-scale ground failure, but solely relying on them can lead to mechanistic misinterpretation.

## Methods

### Surface ejecta change detection

We map earthquake-triggered surface ejecta in Searles and China Lakes using pre- and post-event 5-m PlanetScope OrthoTile (© 2019 Planet Labs PBC) and 10-m Harmonized Sentinel-2 MultiSpectral Instrument Level 2A surface reflectance imagery[54]. Pre-event PlanetScope and Sentinel-2 imagery was acquired on June 29th, 2019, and June 28th, 2019, respectively, and post-event imagery was acquired on 9–10 July 2019 and 8 July 2019, respectively. We use our Sentinel-2-derived change detection results for our overview analysis in Fig. 1, and our PlanetScope-derived change detection results in the hexagon analyses in Figs. 2 and 4.

For both imagery sources, we use the following workflow to identify pixels containing surface ejecta: 1. Band-difference product: subtract each band in the pre-event image from the corresponding band in the post-event image (we use the red, green, and blue bands). 2. Take the absolute value of the band-differenced product. 3. Define a threshold for each band-differenced product and identify pixels that exceed the threshold for all three bands (i.e., red, green, blue). The pixels that exceed the threshold for all three bands are labeled as containing earthquake-triggered surface ejecta. PlanetScope and Sentinel-2 surface reflectance data are unitless, and although the algorithms to produce each product are slightly different, both datasets generally range from 0 to 10,000. We fixed the thresholds for each band based on visual inspection of the data. The threshold was 1100 for all three bands for PlanetScope data, and 100 for the blue band, 500 for the green band, and 300 for the red band for the Sentinel-2 data.

We assume that all above-threshold changes are earthquake-induced surface ejecta because (1) the imagery was acquired in a short timespan (10–11 days) with no other major surface reflectance-altering events (e.g., rainfall) between the pre- and post-event acquisitions; (2) pre-event imagery in the month prior to the earthquake showed little to no change in surface reflectance; and (3) we performed our analysis in low-slope areas where no other major surface changes (e.g., landslides) occurred.

### Optical time series data

We utilized 5-m PlanetScope OrthoTile (© 2019 Planet Labs PBC) data to generate the optical time series depicted in Fig. 2i. All available cloud-free images within the two specified areas of interest were used between February to December 2019, totaling 170 images for the hexagonal ejecta area (Fig. 2a–d) and 185 images for the Zimmaro et al. [1] study area (Fig. 2e–h). Each image is composed of four bands: red, green, blue, and infrared. For a given image, we sampled all four surface reflectance bands at $n$ unmasked pixels. The masking process involved using the surface ejecta change detection layer from the previous section; time series image pixels overlapping with change detection pixels below the specified threshold were considered ejecta-free and consequently masked. The change detection pixels used to mask the time series image pixels are shown in Fig. 2d, h. This sampling approach produced an $n \times 4$ matrix representing $n$ sampled pixels for each of the four optical bands. Next, we calculated the mean ($\mu$) and standard deviation ($\sigma$) for each column, resulting in two $1 \times 4$ vectors, $\mu_v$ and $\sigma_v$. In Fig. 2i, the points correspond to mean($\mu_v$), while the shaded areas depict the range of mean($\sigma_v$).

### InSAR time series methods

We used Sentinel-1a/b imagery[55] from both ascending path 64 and descending path 71 to construct displacement time series covering the

area of interest (Fig. 4d). Data span the time range from 01/02/2015–09/22/2022 with acquisitions every 6–12 days. We processed SAR imagery using the InSAR Scientific Computing Environment[56] (ISCE) and the associated stack processing utility[57]. We used the Shuttle Radar Topography Mission (SRTM) digital elevation model to remove topographic effects[58]. We generated full-resolution interferograms between the nearest two dates for every acquisition in the stack.

2π integer ambiguities were found by unwrapping a filtered version of each interferogram with SNAPHU (Statistical-cost, Network-flow Algorithm for Phase Unwrapping)[59]. Those multiples of 2π were then added to the original interferograms to produce full-resolution, unfiltered, unwrapped interferograms. This approach avoids the potential biases related to phase-closure that result from averaging complex numbers in the presence of asymmetric noise distributions[60].

We inverted our suite of interferograms for the temporal history of displacement at each pixel using the small baseline subset (SBAS) approach[61]. The displacement rate at each pixel was then found using the best-fit linear trend in a least-squares sense for data preceding the Ridgecrest earthquake (01/02/2015–07/04/2019).

Due to the single-dimensionality of InSAR, it is difficult to resolve displacements in standard east-north-up coordinates. Additionally, given the sub north-south flight directions of both ascending and descending paths from Sentinel-1, there is far less resolution in the north-south direction than in the vertical or east-west (E-W) direction. However, with multiple flight paths that provide unique viewing geometries, we can setup an inversion of line-of-sight (LOS) data to approximate displacements in E-W and vertical directions[62]. For example:

$$\widehat{\mathbf{l_j}} = \begin{bmatrix} \sin\left(\theta_j\right)\cos\left(\psi_j\right) \\ \sin\left(\theta_j\right)\sin\left(\psi_j\right) \\ \cos\left(\theta_j\right) \end{bmatrix}. \tag{1}$$

The vector of observations of LOS displacements, $\vec{\mathbf{d}}$ (N × 1) can be related to the E-W and vertical displacement or velocity vector ($\vec{\mathbf{u}}$) in the forward problem

$$\vec{\mathbf{d}} = \mathbf{A}\vec{\mathbf{u}},$$

where the design matrix **A** (N×2), is defined as

$$\mathbf{A} = \begin{bmatrix} \langle \widehat{\mathbf{l_1}}, \widehat{\mathbf{i_1}} \rangle & \langle \widehat{\mathbf{l_1}}, \widehat{\mathbf{i_3}} \rangle \\ \vdots & \vdots \\ \langle \widehat{\mathbf{l_N}}, \widehat{\mathbf{i_1}} \rangle & \langle \widehat{\mathbf{l_N}}, \widehat{\mathbf{i_3}} \rangle \end{bmatrix} \tag{2}$$

When N ≥ 2 and the LOS vectors are not collinear, this problem is not underdetermined, and the values of $\vec{\mathbf{u}}$ can be deduced as the best-fit solution using a linear least-squares approach, $\vec{\mathbf{u}} = \left(\mathbf{A}^T\mathbf{A}\right)^{-1}\mathbf{A}^T\vec{\mathbf{d}}$. This decomposition is known to introduce bias related to the asymmetric nature of the intersection between satellite paths – particularly for southern latitudes[63]. However, in our case, it is likely that the real vertical component of displacement is much larger than the horizontal, therefore the bias is likely insignificant (<5%).

## Data availability

The Planet data are available under restricted access because the U.S. Government contract with Planet does not allow for the publication of raw data in an open repository, access can be obtained by contacting the authors of this study and by reasonable request. The Sentinel-1 data used in this study are available in the Alaska Satellite Facility database https://search.asf.alaska.edu/#/. The Sentinel-2 data used in this study are available in the Google Earth Engine database https://code.earthengine.google.com/. The Cone Penetration Test (CPT) data used in this study are available in the California Geological Survey database by request.

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

## Acknowledgements
We would like to thank Rowena Lohman, Rose Pettiette, Frank Jordan, John Thompson, Sarah Fearkins, and Susan Hall for their advice and expertize. Any use of trade, firm, or product names is for descriptive purposes only and does not imply endorsement by the U.S. Government.

## Author contributions
P.B. led the study, processed initial InSAR data, processed all optical data, and analyzed geologic and geotechnical data. E.M.T. and K.A. provided guidance and key scientific input. K.D.M. processed and analyzed the InSAR data. H.B.M. and S.K.A. provided liquefaction field and data expertize. D.K. provided invaluable location-specific expertize and knowledge.

## Competing interests
The authors declare no competing interests.
