## [Peer Review File · Nature Communications]

The influence of anthropogenic regulation and evaporite dissolution on earthquake-triggered ground failureREVIEWER COMMENTS

Reviewer #1 (Remarks to the Author):

General remarks: The authors present an important case history study regarding the surface ejecta observed in Searles Lake following the 2019 Ridgecrest earthquake. Geologic, geotechnical, and remote sensing data provide evidence that the ejecta was not caused by traditional liquefaction, as previously assumed by past studies. This study points to the influence of solution mining and evaporate dissolution, and hypothesizes a mechanism that likely facilitated the surface manifestations. This case study is a remarkable example of considering alternative datasets and viewpoints to unveil unique lessons. We need more case studies that undertake a similar approach. The article was an enjoyable read and is mostly well-written. My comments suggest improvements in the technical content, figures, and writing.

Technical comments:

- 1) Lines 66-69: The fundamental processes of surface ejecta caused by liquefaction have been well understood and studied for over 50 years. As relevant to this work, I don't believe we need a deeper understanding, but a more comprehensive one. It may be better to say that "we need to holistically consider alternative processes that could lead to surface ejecta" or similar.
- 2) Line 121: Not clear what a "sand-dominated, mud-matrix" is from a geotechnical perspective. Can this be slightly elaborated on? Is there more sand or more "mud" (clays and silts)?
- 3) Line 143-144: Consider revising or removing this definitive statement about anthropogenic regulation. It is well known for example that surface ejecta commonly occurs around the perimeter of building foundations (e.g., shallow footings) over liquefiable soils. Reconnaissance following the 2010-11 Christchurch earthquakes and many others have shown examples of this. I may also consider that an anthropogenic regulation of surface manifestation, as the ejecta finds a pathway to escape along the footing perimeter.
- 4) Line 157-160: The SBT index predicts soil type based on engineering behavior characteristics, and is not a strict predictor of soil composition as implied (e.g., the 2010 version can loosely infer soil consistency and non-uniform soils; the 2016 version used in this work puts more emphasis on critical state soil behavior). Please clarify. Also, gravel as listed is not easily identified from CPT testing and often leads to refusal.
- 5) Line 180: It would be interesting to know the location of this CPT in reference to Fig 2. Is it CPT-1? Can it be indicated in Fig 2?
- 6) Line 182: "Defined above" - I do not see definitions except for in the caption for Figure 3. More details regarding the specific SBT index used should be included here.
- 7) Line 183: The SBT chart in Fig 3 and described here is the 2016 update by Robertson. The citation needs to be corrected for this. Also, this SBT chart is not currently state of practice for liquefaction evaluations. As far as I know, it may not yet be well-conditioned for distinguishing liquefiable soils. One may say that transitional soils and dilative soils from this figure could potentially liquefy. Some discussion as to the reason for selecting this specific chart and details and limitations regarding its interpretations will help clarify the reason for its use here. Alternatively, the 2010 SBT model and related "Ic" index,

coupled with a liquefaction triggering evaluation considering local ground motion parameters (Boulanger and Idriss 2014, or similar) may bring more clarity and basis for rejecting the liquefaction hypothesis. Please consider either further clarifications and/or a more common liquefaction evaluation methodology.

8) Line 200: “Liquefaction” is a fun term. However, there are many mechanisms that fall in a similar gray area. One may argue that the mechanism you describe here could be termed liquefaction: rather than grains contracting you have a cavity collapse, however, it still produces high pore pressures and fluidizes soils such that they eject to surface. For example, static liquefaction, and tsunami or wave-induced liquefaction can also fall under a similar umbrella (high pore pressures occur, but grains may not necessarily contract), but they are still termed liquefaction. You can decide whether to keep this term here, but may be better not to frame it as a “catch-all term” unless you can get into the nuances.

9) Line 215: Others not familiar with InSAR may ask the question: “why were InSAR readings stopped before the earthquake?” Especially because the optical analyses you previously showed does well for before and after the earthquake. As I understand, at least for PSI analyses (I’m not as familiar with SBAS), coherency would be lost during the earthquake both due to larger-scale tectonic movements and local ejecta and subsidence. Can this be briefly explained here for the lay-person?

10) Lines 234-237: Has the ~9 cm subsidence been verified with post-earthquake SBAS / InSAR as well, or just in the field as shown in Figure 5? If this extra information is included, further clarification will be needed.

11) Lines 240-241: I do not see extraction well movements in any of the figures. If this idea is important to your conclusions, which I think it is, an additional subplot to Figure 4 could be added to show extraction well subsidence.

12) Line 271-273: Estimates of the peak ground acceleration or other intensity measure between these two locations could help give a stronger basis for this observation.

13) Line 280: I believe past case histories with similar observations are worthy of elaborating on in slightly greater detail. You can also reference the GEER reconnaissance report for the 2020 Nevada earthquake, where we observed large oddly shaped subsidence features within a salt marsh. I suspect those features formed by a similar mechanism.

Figure comments:

1) ALL: The figure captions are very extensive and provide important details in many instances that should be part of the main text. Please consider moving many of the explanations and figure interpretations to the main body of text.

2) Figure 2 (Line 474-475): Not clear what the “average” standard deviation is here. Perhaps these details should be moved to main text.

3) Figure 3b: Indicate CPT location (perhaps show it on Fig 2 map?)

4) Figure 4d: Yellow point should say “no mining pre-EQ” for consistency with Fig 4c. Consider removing “dv” as this variable is not defined elsewhere.

Editorial comments:

1) Line 32 and throughout: The use of “we” and “our” is used very extensively in this manuscript. I personally try to avoid first-person pronouns in scientific writing, even though passive language may sometimes need to take its place. Please consider revising the text to at least minimize or completely remove first-person pronouns.

- 2) Lines 57 and 199: Replace “heretofore” with “hereafter” or similar
- 3) Line 58: How about “many researchers might assume”? My opinion is that most experienced liquefaction researchers aware of the hexagon pattern would acknowledge it is unusual and would question it rather than just assume it is caused by liquefaction. I suggest the text is generally revised for any other definitive statements like this one.
- 4) Line 104: Not clear on use of “occur.” What about “form” or “becomes visible”?
- 5) Line 107: Instead of “gains,” how about “develops”?
- 6) Line 160-161: Not sure what is meant by “give a full picture.” Please use more descriptive language here.
- 7) Line 164: I do not consider this section as “Observations” as it discusses and evaluates the soil investigation data.
- 8) Line 260 and 275: These headings are not too descriptive, consider revising.

-Patrick Bassal

Reviewer #2 (Remarks to the Author):

This paper takes a look at surface ejecta at Searles Lake observed after the Ridgecrest earthquake. The authors describe a hexagonal shape to the surface ejecta that mimics the pattern of injection wells related to solution mining. Using soil profiles and CPT data, the authors provide evidence that the soil ejecta is not likely the result of liquefaction. The authors then go on to describe an alternative mechanism – cavity collapse resulting in surface ejecta.

The paper is well written and a reasonable short note that responds to prior reporting of liquefaction at Searles Lake. As written, the alternate mechanism for surface ejecta is not well enough developed for publication as a full article. I provide several comments that can be used by the authors to elaborate on their findings and tighten up on their conclusions to bring this paper to publication.

1. Although the title is catchy – I don’t think “liquefiction” is the correct name for the surface ejecta observed at Searles Lake. The surface ejecta is the result of a ground failure and may have a different mechanism than traditional – liquefaction, but should have a reasonable name for future work. It would be better if it was given a name that is more directly related to the cause. Then this paper could be of more use to the community.

2. The liquefaction community often debates the use of CPT data to determine sand-like behavior for liquefaction as CPTs never actually result in a soil sample. Therefore CPT data isn’t the best evidence. If possible – it would be better to lean more heavily on soil samples. It is also true that very thin liquefiable layers can still liquefy and result in surface ejecta, especially when the mining holes are direct conduits to the surface. The authors should be careful with their conclusions – as they are not as definitive as they make them sound. Is there any field data that can relate to the quantity of ejecta at the ejection sites? If the surface ejecta is really just solution that was sitting in the cavity prior to cavity collapse- I would

expect it to be different than typical surface ejecta from liquefaction.

3. Line 143 -144 – what do you mean by direct anthropogenic regulation of the surface manifestation of subsurface ground failure” . I am aware of other efforts such as the blasting experiment with wick drains at Treasure Island and related work on gravel drains: Authors Ashford, Rollins, Lane (2004). Blast-induced liquefaction for full-scale foundation testing. JGGE. Or Mishac Yegian’s work on inducing air bubbles in the subsurface to mitigate liquefaction. There is also bioremediation of soils by DeJong and others to remediate for liquefaction. So be more specific in your meaning. This may be the first example of earthquake-induced cavity collapse subsidence and surface ejecta – but it isn’t the first discussion of anthropogenic impacts on generation of surface effects.

4. Cavity collapse mechanism. If the authors are proposing an alternative mechanism – it would be helpful if it was more fully developed. Figure 6 is helpful as an illustration for cavity collapse – but surface ejecta was also observed away from the injection wells. What is the cavity structure in the region away from the mining operation? Is there any evidence that the quantity of surface ejecta is less away from injection wells – or is it the same? The InSAR demonstrates that the subsidence is less. It makes sense to me that the injection wells provide an easy path for water escape – so I would expect more surface eject here than where there are no wells? Is there evidence for that?

5. Figure 4c. the yellow color does not show ejecta – it shows an area without ejecta. Change the label.

6. The InSAR results are not well integrated into this work. They are potentially interesting on their own – but don’t seem to tie in well to the main argument of the paper. I believe Figure 4 shows that there is more subsidence when there is solution mining. How does that help us determine the mechanism for surface ejecta? That connection is not well made. As it currently reads – it almost seems like a parallel result.

Reconciliation of Reviews for “Liquefaction or liquefaction? The influence of anthropogenic regulation and evaporite dissolution on earthquake-triggered ground failure”

We thank the two reviewers for their very thorough and constructive commentary and suggestions. We have addressed their concerns and describe our changes on a point-by-point basis below. The main changes to the manuscript are as follows: (1) an overall softening of our language to allow for uncertainty and flexibility in our conclusions, (2) a supplemental information document, which includes several additional CPT analyses and a more extensive description of the field observed versus InSAR-derived injection well subsidence, (3) a more nuanced discussion of liquefaction and the cavity collapse mechanism proposed here, and (4) additional subsections that delve more deeply into other potential cavity collapse features, both in Searles Lake and in other earthquake events that triggered similar ground failure.

Reviewer #1:

General remarks: The authors present an important case history study regarding the surface ejecta observed in Searles Lake following the 2019 Ridgecrest earthquake. Geologic, geotechnical, and remote sensing data provide evidence that the ejecta was not caused by traditional liquefaction, as previously assumed by past studies. This study points to the influence of solution mining and evaporite dissolution, and hypothesizes a mechanism that likely facilitated the surface manifestations. This case study is a remarkable example of considering alternative datasets and viewpoints to unveil unique lessons. We need more case studies that undertake a similar approach. The article was an enjoyable read and is mostly well-written. My comments suggest improvements in the technical content, figures, and writing.

Technical comments:

1) Lines 66-69: The fundamental processes of surface ejecta caused by liquefaction have been well understood and studied for over 50 years. As relevant to this work, I don't believe we need a deeper understanding, but a more comprehensive one. It may be better to say that “we need to holistically consider alternative processes that could lead to surface ejecta” or similar.

Change made.

2) Line 121: Not clear what a “sand-dominated, mud-matrix” is from a geotechnical perspective. Can this be slightly elaborated on? Is there more sand or more “mud” (clays and silts)?

We meant that there are sand layers thick enough to lead to earthquake-induced surface ejecta within an otherwise mud-dominated soil profile. To address the comment, we removed “mud-matrix” from the sentence.

3) Line 143-144: Consider revising or removing this definitive statement about anthropogenic regulation. It is well known for example that surface ejecta commonly occurs around the perimeter of building foundations (e.g., shallow footings) over liquefiable soils. Reconnaissance following the 2010-11 Christchurch earthquakes and many others have shown examples of this. I may also consider that an anthropogenic regulation of surface manifestation, as the ejecta finds a pathway to escape along the footing perimeter.

The authors agree, and we've softened the second clause of the sentence to read "which provides more documented evidence of direct anthropogenic regulation of the surface manifestation of subsurface ground failure."

4) Line 157-160: The SBT index predicts soil type based on engineering behavior characteristics, and is not a strict predictor of soil composition as implied (e.g., the 2010 version can loosely infer soil consistency and non-uniform soils; the 2016 version used in this work puts more emphasis on critical state soil behavior). Please clarify. Also, gravel as listed is not easily identified from CPT testing and often leads to refusal.

We appreciate your technical comment 7, which is directly related. Our response to that comment should also satisfy this comment. We have also removed "gravel", as we agree that it is not easily identified from CPT testing.

5) Line 180: It would be interesting to know the location of this CPT in reference to Fig 2. Is it CPT-1? Can it be indicated in Fig 2?

The location of the CPT is given in Fig. 1 as CPT 1, as you've correctly noted. It's beyond the extent of the maps for both areas displayed in Fig. 2, which is why it's not plotted there. We have added the following sentence to the Fig. 1 caption to make the location clearer, "Note that "CPT 1" in (b) is the location of the cone penetration test sounding shown in Fig. 3." We've also added a note that the images in first row of Fig. 2 have the same footprint as the images in Fig. 1c-d, which adds location context. The Fig. 3 caption already directs readers back to Fig. 1b for the CPT location.

6) Line 182: "Defined above" - I do not see definitions except for in the caption for Figure 3. More details regarding the specific SBT index used should be included here.

The parenthetical "defined above" is a relic from a previous draft that we shortened for page limit considerations. We have removed it. The reader can see the soil-behavior types in the Fig. 3 caption and supplemental material.

7) Line 183: The SBT chart in Fig 3 and described here is the 2016 update by Robertson. The citation needs to be corrected for this. Also, this SBT chart is not currently state of practice for liquefaction evaluations. As far as I know, it may not yet be well-conditioned for distinguishing liquefiable soils. One may say that transitional soils and dilative soils from this figure could potentially liquefy. Some discussion as to the reason for selecting this specific chart and details and limitations regarding its interpretations will help clarify the reason for its use here. Alternatively, the 2010 SBT model and related “ I_c ” index, coupled with a liquefaction triggering evaluation considering local ground motion parameters (Boulanger and Idriss 2014, or similar) may bring more clarity and basis for rejecting the liquefaction hypothesis. Please consider either further clarifications and/or a more common liquefaction evaluation methodology.

The citation has been updated. Furthermore, following your review comment, we have performed additional analyses to incorporate the Robertson (2010) soil behavior index, I_c index, liquefaction potential index (LPI) computations, and probability of surface manifestation of liquefaction. We include the $I_c=2.6$ boundary in Fig. 2 and a discussion of the percentages of data that fall above and below this boundary in the main body of the paper. The additional analyses did not change the major conclusions of the paper and were too extensive to fit in the main body of the text. Thus, we created a supplemental document to include these analyses.

8) Line 200: “Liquefiction” is a fun term. However, there are many mechanisms that fall in a similar gray area. One may argue that the mechanism you describe here could be termed liquefaction: rather than grains contracting you have a cavity collapse, however, it still produces high pore pressures and fluidizes soils such that they eject to surface. For example, static liquefaction, and tsunami or wave-induced liquefaction can also fall under a similar umbrella (high pore pressures occur, but grains may not necessarily contract), but they are still termed liquefaction. You can decide whether to keep this term here, but may be better not to frame it as a “catch-all term” unless you can get into the nuances.

These are excellent points, and we agree on all counts. To begin, we have inserted an explicit definition of what we term “classical liquefaction”, so we have a specified mechanism that we can compare our proposed cavity collapse mechanism against. This has been added to the beginning of the “Was liquefaction responsible for the hexagonal ejecta?” section. At the end of this section, after we conclude that the main mechanism driving the hexagonal ejecta is likely not “classical liquefaction”, we discuss what we mean by “liquefiction”. Rather than attempting to define liquefaction as a “catch all” mechanistic term, we have decided to use it to describe the misinterpretation of earthquake-triggered ground failure observations as classical liquefaction-driven. In this way, we do not use “liquefiction” to describe the mechanism responsible for the hexagonal ejecta, but rather a way to describe the misinterpretation of the surface ejecta. As part of addressing this comment, as well as comment 1 from reviewer 2, we have also explicitly introduced the term “dissolution cavity collapse” or “DCC” as the dominant mechanism that drove the hexagonal surface ejecta. We hope that this further disaggregates our use of “liquefiction” from an actual defined mechanistic term, DCC. Finally, we greatly

appreciated your example of wave/tsunami-induced liquefaction. Based on that, we have added a more nuanced discussion of the DCC mechanism as a potential sub-type of liquefaction. We have kept this section concise but hope that it addresses subtleties that our first draft did not.

9) Line 215: Others not familiar with InSAR may ask the question: "why were InSAR readings stopped before the earthquake?" Especially because the optical analyses you previously showed does well for before and after the earthquake. As I understand, at least for PSI analyses (I'm not as familiar with SBAS), coherency would be lost during the earthquake both due to larger-scale tectonic movements and local ejecta and subsidence. Can this be briefly explained here for the lay-person?

We have added an explanatory sentence to the InSAR section explaining why we cannot use InSAR data in this location spanning the earthquake (your reasoning was exactly right!).

10) Lines 234-237: Has the ~9 cm subsidence been verified with post-earthquake SBAS / InSAR as well, or just in the field as shown in Figure 5? If this extra information is included, further clarification will be needed.

To address this comment, we have added a supplementary section that includes an InSAR time series sampled at this pixel location. We have decided to combine this figure with Fig. 5, and together, the photos of the injection well and InSAR time series have been moved to Fig. S2. The reason for this is because the field observation and related InSAR time series require significant explanation that is not critical to the main argument of the paper, since it is only used to confirm that some degree of subsidence is observed at injection wells via two independent methods (i.e., InSAR and field observation).

11) Lines 240-241: I do not see extraction well movements in any of the figures. If this idea is important to your conclusions, which I think it is, an additional subplot to Figure 4 could be added to show extraction well subsidence.

We have re-worked Fig. 4 to help enhance clarity and have also added an additional subplot to Fig. 4 that include the movements at the extraction wells associated with each mining epoch (i.e., mining beginning pre-2015 and mid-2017).

12) Line 271-273: Estimates of the peak ground acceleration or other intensity measure between these two locations could help give a stronger basis for this observation.

We have added the peak ground acceleration based on the USGS ShakeMap model for the two core locations in China Lake and Searles Lake.

13) Line 280: I believe past case histories with similar observations are worthy of elaborating on in slightly greater detail. You can also reference the GEER reconnaissance report for the 2020 Nevada earthquake, where we observed large oddly shaped subsidence features within a salt marsh. I suspect those features formed by a similar mechanism.

We appreciate the reference and have added a paragraph dedicated to the observations from the 2020 Monte Cristo, Nevada earthquake with reference to the GEER report. We have also elaborated on another example from the 2001 Bhuj earthquake, and more explicitly stated the importance of these analogs.

Figure comments:

1) ALL: The figure captions are very extensive and provide important details in many instances that should be part of the main text. Please consider moving many of the explanations and figure interpretations to the main body of text.

We appreciate the comment and have removed some text from the longer figure captions (i.e., Figs 3 and 4). However, USGS guidelines require us to elaborate on captions such that they can be read as standalone items without body text to understand the figure. Also, given the variety of topics in the paper and the larger target audience, the figure captions have not been substantially shortened.

2) Figure 2 (Line 474-475): Not clear what the “average” standard deviation is here. Perhaps these details should be moved to main text.

Rather than adding this information to the main text, we have added a full description of our methods in generating Fig. 2i to the *Methods* section. We have also removed the methodology explanation from the Fig. 2 caption, and instead added reference to the *Methods* section.

3) Figure 3b: Indicate CPT location (perhaps show it on Fig 2 map?)

We addressed this comment with your technical comment 5.

4) Figure 4d: Yellow point should say “no mining pre-EQ” for consistency with Fig 4c. Consider removing “dv” as this variable is not defined elsewhere.

We have re-worked this figure, and hope that the changes have clarified this point.

Editorial comments:

1) Line 32 and throughout: The use of “we” and “our” is used very extensively in this manuscript. I personally try to avoid first-person pronouns in scientific writing, even though passive language may sometimes need to take its place. Please consider revising the text to at least minimize or completely remove first-person pronouns.

We respect the reviewers scientific writing style and understand that our writing style may be seen as non-traditional. However, the geological sciences disciplines allow for more first-person usage than the engineering disciplines. Furthermore, the work presented in this paper did not follow a traditional “question-results-discussion” structure that lends itself to a third person writing style. Rather, it was an exploration of a unique observation that led the authors from one result to the next, and we believe that writing the paper as such lends itself much more to a first-person style. Given this and our larger target audience, we have respectfully decided to keep our style.

2) Lines 57 and 199: Replace “heretofore” with “hereafter” or similar

First instance replaced with “heretofore” and second instance replaced with “hereafter.”

3) Line 58: How about “many researchers might assume”? My opinion is that most experienced liquefaction researchers aware of the hexagon pattern would acknowledge it is unusual and would question it rather than just assume it is caused by liquefaction. I suggest the text is generally revised for any other definitive statements like this one.

We deleted this entire sentence to incorporate another reviewer’s comment. We also scoured the manuscript in general during our thorough rereview to soften some of the language.

4) Line 104: Not clear on use of “occur.” What about “form” or “becomes visible”?

We have changed “occur” to “become visible.”

5) Line 107: Instead of “gains,” how about “develops”?

Change made.

6) Line 160-161: Not sure what is meant by “give a full picture.” Please use more descriptive language here.

We have revised this sentence entirely, and hope it provides a more descriptive explanation of our meaning: “Here, we review the available geologic and geotechnical data to understand the

mechanism that resulted in surface ejecta, given an initial expectation that surface ejecta typically results from liquefaction”.

7) Line 164: I do not consider this section as “Observations” as it discusses and evaluates the soil investigation data.

We agree with the reviewer that this usage of “Observations” is incorrect. We have revised to “Geologic analysis” and “Geotechnical analysis”.

8) Line 260 and 275: These headings are not too descriptive, consider revising.

Thank you for this comment. After our extensive re-working of the main text, we have changed many of the section titles. We believe that these changes are more descriptive.

Reviewer #2

This paper takes a look at surface ejecta at Searles Lake observed after the Ridgecrest earthquake. The authors describe a hexagonal shape to the surface ejecta that mimics the pattern of injection wells related to solution mining. Using soil profiles and CPT data, the authors provide evidence that the soil ejecta is not likely the result of liquefaction. The authors then go on to describe an alternative mechanism – cavity collapse resulting in surface ejecta.

The paper is well written and a reasonable short note that responds to prior reporting of liquefaction at Searles Lake. As written, the alternate mechanism for surface ejecta is not well enough developed for publication as a full article. I provide several comments that can be used by the authors to elaborate on their findings and tighten up on their conclusions to bring this paper to publication.

1. Although the title is catchy – I don’t think “liquefaction” is the correct name for the surface ejecta observed at Searles Lake. The surface ejecta is the result of a ground failure and may have a different mechanism than traditional – liquefaction, but should have a reasonable name for future work. It would be better if it was given a name that is more directly related to the cause. Then this paper could be of more use to the community.

We agree with the reviewer and have also addressed a similar comment by reviewer 1 (comment # 8). For a full description of the changes we have made, please see our response to that comment. In response to this specific issue, we have formally named the mechanism described in this paper as “dissolution cavity collapse”, or “DCC”, which can be used by the community in future work.

2. The liquefaction community often debates the use of CPT data to determine sand-like behavior for liquefaction as CPTs never actually result in a soil sample. Therefore CPT data isn't the best evidence. If possible – it would be better to lean more heavily on soil samples. It is also true that very thin liquefiable layers can still liquefy and result in surface ejecta, especially when the mining holes are direct conduits to the surface. The authors should be careful with their conclusions – as they are not as definitive as they make them sound. Is there any field data that can relate to the quantity of ejecta at the ejection sites? If the surface ejecta is really just solution that was sitting in the cavity prior to cavity collapse- I would expect it to be different than typical surface ejecta from liquefaction.

We thank the reviewer for this comment and have addressed it in the following ways:

1. We have softened the language around the conclusions we draw from the CPT data, and have softened the language throughout the text to assure that our conclusions are not as definitive as they originally sounded.

2. We have added additional CPT data analysis to provide other perspectives, as suggested in a reviewer 1 comment. As part of this, we include an analysis of the “severity” of ejecta using an analysis in Geyin and Maurer (2020). Because we do not have any more field observations than those already presented, we use the Geyin and Maurer (2020) analysis to address the reviewer’s comment regarding the expectation and observation of the quantity of ejecta. We believe that this suffices.

3. We have included the reviewer’s point that there still may exist very thin liquefiable layers under the hexagonal ejecta and cannot fully reject that classical liquefaction was not at all responsible for the observation. However, based on our second response to this comment, we maintain that the extensiveness and scale of the ejecta features is unlikely to have occurred due to liquefaction alone.

4. We agree that one would expect a different type of surface ejecta from DCC than classical liquefaction, and we believe that this is reflected in the unusually large spatial scale of the individual ejection features and also the evolution of the surface reflectance in the months following the earthquake, where the DCC-triggered ejecta took on the white hue indicative of dried, evaporite-rich fluids, whereas the sand boil features identified by Zimmaro et al. (2020) did not. We have added a paragraph in the “surface ejecta observations” section to elaborate on the latter point.

3. Line 143 -144 – what do you mean by direct anthropogenic regulation of the surface manifestation of subsurface ground failure” . I am aware of other efforts such as the blasting experiment with wick drains at Treasure Island and related work on gravel drains: Authors Ashford, Rollins, Lane (2004). Blast-induced liquefaction for full-scale foundation testing. JGGE. Or Mishac Yegian’s work on inducing air bubbles in the subsurface to mitigate liquefaction. There is also bioremediation of soils by DeJong and others to remediate for liquefaction. So be more specific in your meaning. This may be the first example of earthquake-induced cavity collapse subsidence and surface ejecta – but it isn't the first discussion of anthropogenic impacts on generation of surface effects.

We agree with the reviewer and acknowledge that our language was incorrect. We have changed the statement “we believe this is the first documented evidence” to “which provides more documented evidence”.

4. Cavity collapse mechanism. If the authors are proposing an alternative mechanism – it would be helpful if it was more fully developed. Figure 6 is helpful as an illustration for cavity collapse – but surface ejecta was also observed away from the injection wells. What is the cavity structure in the region away from the mining operation? Is there any evidence that the quantity of surface ejecta is less away from injection wells – or is it the same? The InSAR demonstrates that the subsidence is less. It makes sense to me that the injection wells provide an easy path for water escape – so I would expect more surface eject here than where there are no wells? Is there evidence for that?

We thank the reviewer for this comment. Prior to our first submission, we had given this point a lot of consideration, and we had decided that discussion of other ejecta features in Searles Lake was out of the scope of this study. While we still believe a full analysis of other Searles Lake ejecta features would distract from the focus of the paper, we have added a sub-section called “within Searles lake” that directly addresses question of surface ejecta observed away from the solution mining injection wells by focusing on one representative ejecta feature. In this section, we have also included several new references that substantiate our claim of natural rainwater percolation as a viable mechanism to create dissolution cavities. While we could not go into full detail about the potential differences in cavity structure, extent of ejecta, etc. due to word limitations, we believe this section helps more fully develop the DCC mechanism. In addition, our new sub-section “other earthquakes” also helps more fully develop our proposed DCC mechanism.

5. Figure 4c. the yellow color does not show ejecta – it shows an area without ejecta. Change the label.

We have re-worked this figure, and our changes have addressed this point.

6. The InSAR results are not well integrated into this work. They are potentially interesting on their own – but don’t seem to tie in well to the main argument of the paper. I believe Figure 4 shows that there is more subsidence when there is solution mining. How does that help us determine the mechanism for surface ejecta? That connection is not well made. As it currently reads – it almost seems like a parallel result.

To address this comment, we have added text to the abstract, alternative mechanism section, and conclusions that explicitly explain why the InSAR data are an important tool in understanding the DCC mechanism. In sum, the InSAR data are the only evidence we have that

supports our theory that the solution mining is indeed causing substantial volume loss that would lead to the generation of subsurface cavities.

REVIEWERS' COMMENTS

Reviewer #2 (Remarks to the Author):

The authors have appropriately addressed my review comments. I believe the manuscript is ready for publication. This is a well written manuscript that will be interesting to the liquefaction community.